# New climate models reveal faster and larger increases in Arctic precipitation than previously projected

Michelle R. McCrystall [1✉], Julienne Stroeve[1,2,3], Mark Serreze[3], Bruce C. Forbes[4] & James A. Screen [5]

As the Arctic continues to warm faster than the rest of the planet, evidence mounts that the region is experiencing unprecedented environmental change. The hydrological cycle is projected to intensify throughout the twenty-first century, with increased evaporation from expanding open water areas and more precipitation. The latest projections from the sixth phase of the Coupled Model Intercomparison Project (CMIP6) point to more rapid Arctic warming and sea-ice loss by the year 2100 than in previous projections, and consequently, larger and faster changes in the hydrological cycle. Arctic precipitation (rainfall) increases more rapidly in CMIP6 than in CMIP5 due to greater global warming and poleward moisture transport, greater Arctic amplification and sea-ice loss and increased sensitivity of precipitation to Arctic warming. The transition from a snow- to rain-dominated Arctic in the summer and autumn is projected to occur decades earlier and at a lower level of global warming, potentially under 1.5 °C, with profound climatic, ecosystem and socio-economic impacts.

[1] Canada 150 Research Chairs Program, Centre for Earth Observation Science, University of Manitoba, Winnipeg, MB, Canada. [2] Department of Earth Sciences, University College London, London, UK. [3] National Snow and Ice Data Centre, Cooperative Institute for Research in Environmental Sciences, University of Colorado Boulder, Boulder, Colorado, USA. [4] Arctic Centre, University of Lapland, Rovaniemi, Finland. [5] College of Engineering, Maths, and Physical Sciences, University of Exeter, Exeter, UK. ✉email: michelle.mccrystall@umanitoba.ca

There is general agreement that Arctic precipitation will increase through the twenty-first century, with estimates ranging from 30% to 60% by the year 2100[1–3]. A wetter Arctic results from [1] increased evaporation as a result of more open water due to sea-ice loss[1,4,5]; [2] higher air temperatures, increasing the atmosphere's ability to carry moisture[6–8]; and [3] increased poleward moisture transport[9–11].

The Arctic is also expected to transition from a largely snow-dominated to a rain-dominated precipitation regime[3], a transition already being observed over the Atlantic sector[12]. However, uncertainty exists regarding the regional extent and seasonality of these changes. Previous studies conclude that rainfall will increase in spring, autumn[2] and winter[3], whereas rainfall and snowfall are projected to increase over some regions during autumn and winter[2,13]. This increased, rainfall-dominated precipitation could have pronounced impacts on Greenland ice sheet mass balance and global sea level[14,15], river discharge[11,16], Arctic sea-ice extent and thickness[17], permafrost[18], as well as flora, fauna and linked social-ecological systems[18–21].

Outputs from the new Coupled Model Intercomparison Project Phase 6 (CMIP6) experiments provide an opportunity to assess the latest projections of climate change under various emissions forcing scenarios. Relative to CMIP5, CMIP6 has improved simulations of the sea-ice mean state and trends over the period of satellite observations[22,23], as well as improved simulations of historical snow cover[8] and global precipitation intensities[24]. This suggests that other aspects of the hydrological cycle, such as Arctic precipitation, are also improved.

Here we examine projections of Arctic precipitation change through 2100 from CMIP6 and compare these to those from the earlier CMIP5. The key conclusion is that CMIP6 projects larger and faster increases in precipitation and an earlier transition to a rainfall-dominated Arctic in the summer and autumn.

## Results

**Historical changes.** We first assess the robustness of simulated precipitation from both CMIP5 and CMIP6 against two Arctic precipitation data sources: the ERA5 atmospheric reanalysis[25] and the Global Precipitation Climatology Project (GPCP) product[26], which merges data from gauges (that have sparse coverage in the Arctic and are known to suffer from under-catch of snowfall), and satellites. Relative to the GPCP analysis, both CMIP5 and CMIP6 simulate, on average, 0.04–0.1 mm day$^{-1}$ more Arctic (70–90°N) precipitation over the historical time period (1979–2005) (Fig. 1a). However, the veracity of GPCP data is questionable[27]. Marcovecchio et al.[28] found that GPCP had lower precipitation rates and different seasonal variations compared to reanalyses. It is noteworthy from Fig. 1 that interannual variations between GPCP and ERA5 are not strongly correlated ($r = 0.33$). To offer two examples, low precipitation in 2000 and high precipitation in 2005 from the GPCP product are not seen in ERA5. However, the annually averaged spatial precipitation patterns in GPCP (Fig. 1c) are similar to ERA5 (Fig. 1e), albeit of weaker magnitude. In line with Marcovecchio et al.[28], this suggests a potential low bias in GPCP. Although ERA5 is a reanalysis product, it has been shown to have good representation of precipitation over the Arctic relative to buoy and satellite data, and is superior to its predecessor, ERA-Interim. Two recent studies conclude that ERA5 is the best data set currently available to represent precipitation across the Arctic region[29,30].

Annual-mean Arctic precipitation in both CMIP5 and CMIP6 is consistent with ERA5, with values of around 0.94 ± 0.03 mm day$^{-1}$ for each data set. Further, both CMIP5 and CMIP6 multi-model

ensemble means simulate similar spatial patterns of precipitation variability as ERA5 (Fig. 1d–f). Overall, the CMIP5 multi-model ensemble mean (37 models) of annual precipitation is larger by ~0.1 ± 0.04 mm day$^{-1}$ than CMIP6 (32 models) over the same historical time period (Fig. 1a). We also compare the snowfall-to-precipitation ratio (snowfall divided by total precipitation; hereafter simply referred to as the snowfall ratio) between ERA5, CMIP5 and CMIP6 (Fig. 1b). Throughout the observational record (1979–2005), there is a slight statistically significant negative trend in the snowfall ratio in ERA5, which is captured by both CMIP ensembles, resulting in a reduction of around −0.06 across the observational record. In most years, ERA5 falls within the spread of CMIP6 but not CMIP5, suggesting an improvement in the partitioning between rain and snow in CMIP6.

**End-of-century model projected changes.** The multi-model ensemble mean of Arctic precipitation increases in all seasons throughout the twenty-first century, especially in autumn, for the Representative Concentration Pathway 8.5 (RCP8.5)/Shared Socioeconomic Pathway 5–8.5 (SSP5–8.5) scenarios in CMIP5 and CMIP6, respectively (Fig. 2). This total precipitation increase is largely dominated by an increase in rainfall for all seasons in both CMIP ensembles. In summer and autumn, the rainfall increase is accompanied by decreased snowfall. In winter, however, snowfall continues to increase and remains the dominant precipitation type at the end of the century across most of the Arctic (see Fig. 8). In spring, there is little change in snowfall throughout the century.

Overall, CMIP6 projects larger increases in precipitation than in CMIP5, dominated by increased rainfall. At the end of the century (2100) relative to the year 2000, there is a 422% increase in CMIP6 rainfall compared to 260% in CMIP5 in winter; corresponding values are 261% and 141% in spring, 71% and 51% in summer, and 268% and 192% in autumn. This results in ~0.3 mm day$^{-1}$ or 27.3 mm per season, difference in rainfall by 2100 between the two CMIPs in autumn and around a 0.2 mm day$^{-1}$ (18.2 mm per season) difference in spring and winter. Trends are also larger in CMIP6—in autumn, rainfall increases by 0.9 mm day$^{-1}$ (81.9 mm per season) from 2020 to 2100 compared to 0.7 mm day$^{-1}$ (63.7 mm per season) in CMIP5, resulting in a 24% larger rainfall increase. Larger rainfall increases are also simulated in other seasons—winter has a 39% greater increase in CMIP6, whereas spring and summer have 36% and 14% greater increases, respectively. There is a greater reduction of snowfall in summer (16%) and autumn (38%) at the end of the century in CMIP6, consistent with a shorter snow cover season than previously simulated[8]. Similar patterns emerge for the RCP4.5 scenario (Supplementary Fig. 1), although the precipitation change per season is more modest than for RCP8.5.

Not only are the multi-model ensemble mean changes larger in CMIP6 than CMIP5, there is also a larger inter-model spread in CMIP6, indicating greater uncertainty in projected precipitation changes (Supplementary Table 1). The greater spread in total precipitation, rainfall and snowfall in CMIP6 is likely related to the greater spread in surface air temperature, open water and vertically integrated moisture flux (VIMF) at the end of the century (2091–2100) (Supplementary Tables 1 and 2). Accompanying this larger spread is an increase in the upper range, with the upper 95th percentile in autumn at 1.41 mm day$^{-1}$ in CMIP6 compared to 1.26 mm day$^{-1}$ in CMIP5 (Supplementary Table 1), suggesting that greater average rainfall amounts are possible by 2100 than previously projected. For example, in autumn, the upper limit of the CMIP6 projections is 1.7 mm day$^{-1}$ (154.7 mm

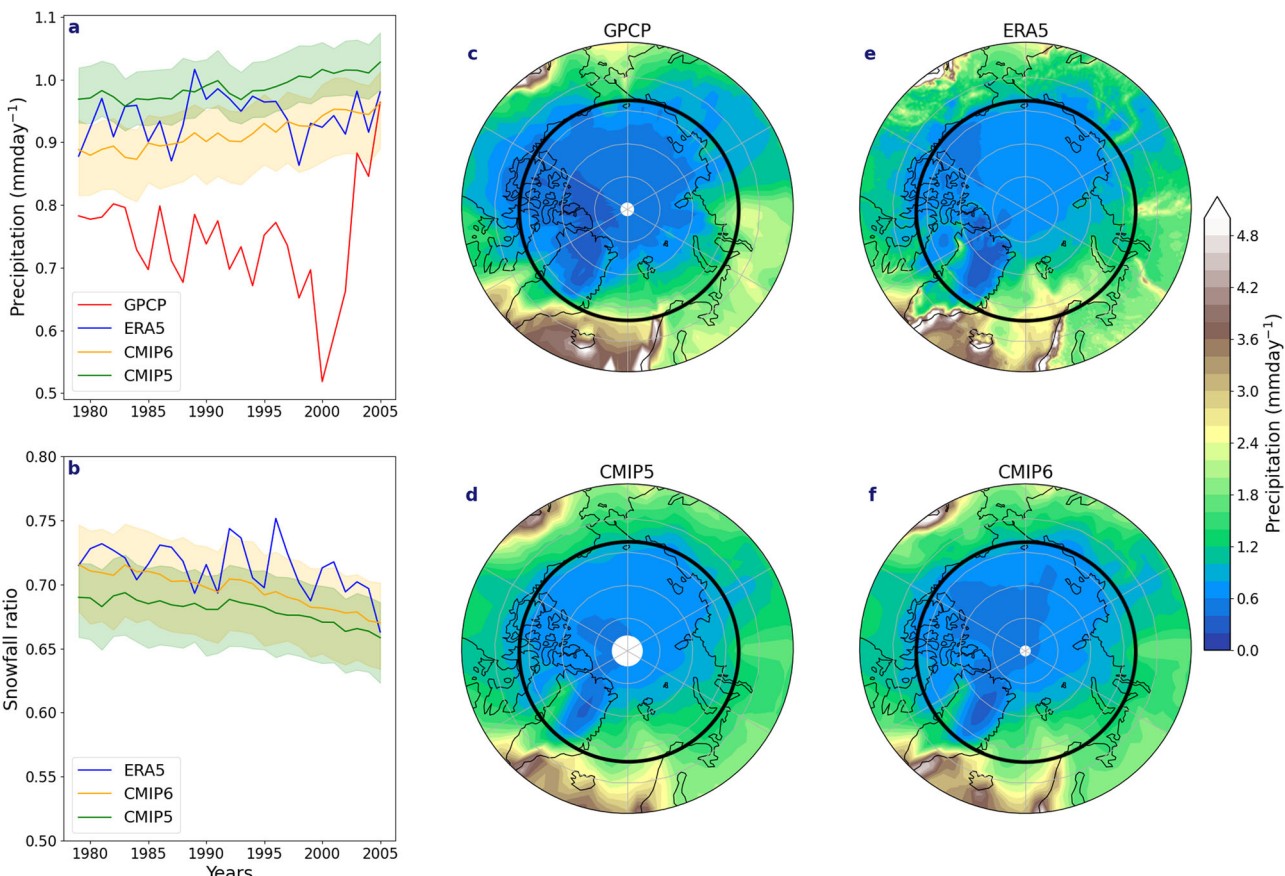

**Fig. 1 Time series of precipitation, and snowfall ratio and spatial climatology of total precipitation. a** Annual precipitation means over the Arctic (70°N–90°N) from 1979 to 2005, covering the satellite era and the start of the GPCP record concluding with the end of the historical period for GPCP, ERA5, CMIP5 and CMIP6. **b** Annual snowfall ratio (snowfall/total precipitation) over the Arctic for the same observational record of 1979–2005 for ERA5, CMIP6 and CMIP5. The shading in both represents the 5th to 95th percentile spread in the models. Climatological precipitation for the same record over the Arctic for **c** GPCP, **d** CMIP5, **e** ERA5 and **f** CMIP6. The Arctic region for all time series analyses is taken as the area poleward of 70°N, shown as the dark circle in **c**–**f**.

per season) compared to 1.4 mm day⁻¹ (127.4 mm per season) in CMIP5.

More extensive spatial changes in snowfall and rainfall across the Arctic are also projected in CMIP6, especially in autumn and winter under RCP8.5 (Fig. 3) and to a lesser degree in RCP4.5 (Supplementary Fig. 2). By 2100, large increases in autumn rainfall are simulated across the Arctic in both CMIP ensembles (Fig. 3g) but are more pronounced in CMIP6, with statistically significant increases of up to 0.6 mm day⁻¹ around the Greenland and Barents Seas (Fig. 3h). This is coupled with a larger statistically significant decrease in autumn snowfall of around 0.4 mm day⁻¹ compared to CMIP5 (Fig. 3f), except in East Greenland. Although differences between CMIP6 and CMIP5 are spatially more limited in winter relative to autumn (Fig. 3b), larger increases in rainfall are apparent in CMIP6 across the Arctic Ocean and peripheral seas, especially the Greenland and Barents Seas (Fig. 3d). Significant snowfall increases are projected by 2100 in winter (Fig. 3a) consistent with findings from Krasting et al.[31] in their assessment of CMIP5 simulations. However, these snowfall increases are again greater in CMIP6, especially in Siberia and the Canadian Arctic Archipelago. Although differences between CMIP6 and CMIP5 in spring and summer are less pronounced (Supplementary Fig. 3), CMIP6 simulates a larger increase in rainfall (Supplementary Fig. 3d, h) coinciding with a larger snowfall reduction relative to CMIP5, throughout the Arctic and in both seasons.

**Drivers of precipitation change**. Stronger Arctic precipitation increases in CMIP6 could be the result of either a greater sensitivity of precipitation change per degree of warming (defined here as the percentage change in precipitation per degree of Arctic or global warming) or more pronounced warming in CMIP6 models[32]. During autumn and early winter, CMIP6 exhibits a stronger precipitation increase per degree of Arctic (Fig. 4a) and global warming (Fig. 4b), than in CMIP5, although the difference between CMIP5 and CMIP6 is smaller for the sensitivity to Arctic warming (Fig. 4a). The relatively larger difference in the precipitation sensitivity to global warming than Arctic warming, in CMIP6 compared to CMIP5, reflects the larger Arctic amplification of warming in CMIP6[33]. Slightly larger precipitation sensitivity to warming during the freeze-up season may in part explain the greater change in projected autumn precipitation.

In both CMIP ensembles, the magnitude of rainfall increase (and snowfall decrease) at the end of the century is statistically significantly correlated (at the 95% confidence level, with *r*-values between 0.6 and 0.89) with the magnitude of Arctic warming in all seasons (Fig. 5a, d, g, j and Supplementary Table 3). Enhanced Arctic warming in CMIP6 is most pronounced in winter, when the CMIP6 multi-model ensemble mean is around 15 °C by 2100 (relative to the start of the century), compared to 13 °C in CMIP5 (Fig. 5a, d, g, j and Fig. 6b). However, the spread between individual models is large, with the maximum value from one CMIP6 model ensemble projecting up to 23 °C warming at the

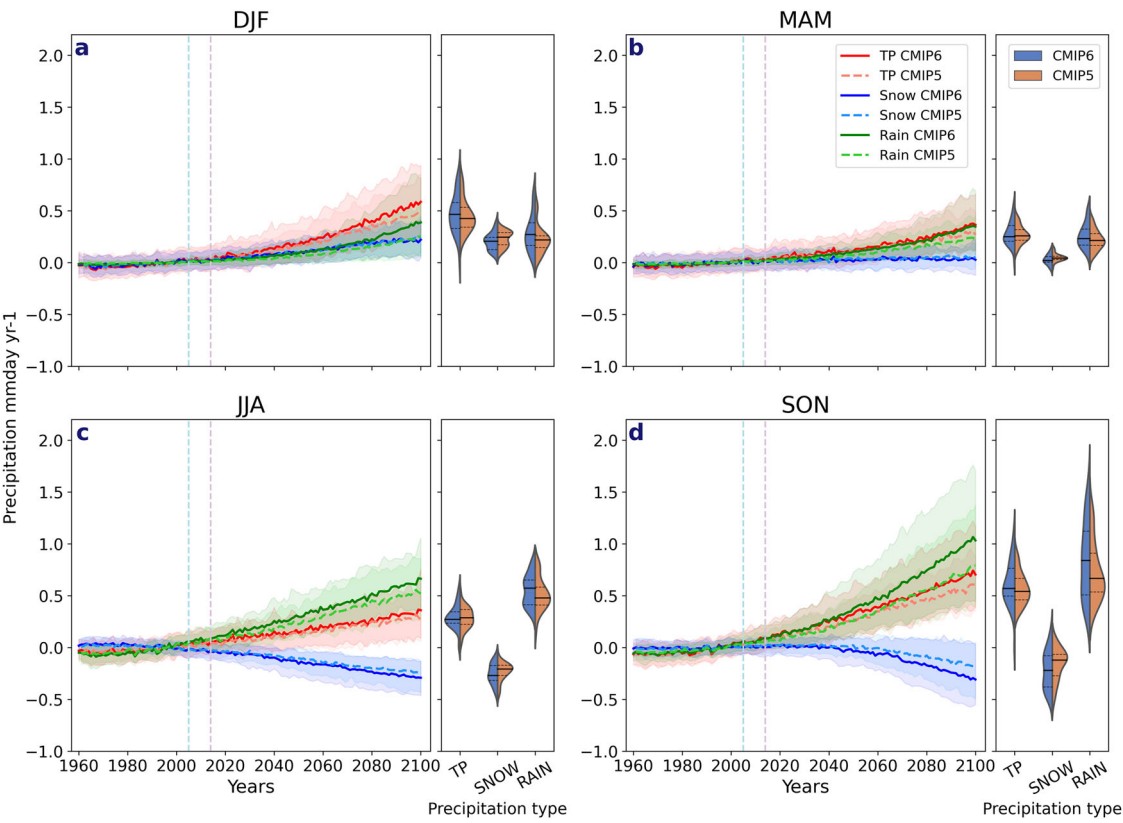

**Fig. 2 Time series of total precipitation, snowfall and rainfall anomalies.** Changes in total precipitation (TP) (red, orange), snowfall (snow) (blue, light blue) and rainfall (rain) (green, light green) in CMIP6 and CMIP5 are shown relative to the 1981–2009 climatological mean for **a** December–February (DJF), **b** March–May (MAM), **c** June–August (JJA) and **d** September–November (SON). The light blue vertical dashed line denotes when the historical period for CMIP5 ends and the light purple vertical dashed line denotes when the historical period of CMIP6 ends and thereafter the RCP8.5 and SSP5–8.5 climate scenarios for CMIP5 and CMIP6 are used. The shading around each line highlights the spread based upon the lower 5th and 95th percentiles among the model members. The violin plots represent the model spread from 2090 to 2100 for each total precipitation (TP), snowfall (snow) and rainfall (rain) with the dashed black lines representing the 25th and 75th percentiles, and the black vertical line representing the mean of all models.

end of the century compared to up to 18 °C warming in an individual model in the CMIP5 ensemble. Other seasons are characterized by more modest increases, with a smaller spread between models, and smaller differences between CMIP6 and CMIP5. Nonetheless, even with these more moderate temperature increases in other seasons and between CMIP ensembles, CMIP6 does show a greater magnitude of rainfall increase across the models with the amount of Arctic warming (Fig. 5). We hence partly attribute the differences in precipitation projections in CMIP5 and CMIP6 to stronger projected increases in global and Arctic air temperature in CMIP6. However, larger magnitude global warming is insufficient to explain the faster projected precipitation changes in CMIP6, particularly in autumn, as we find that for the specific levels of global warming (1.5 °C, 2 °C and 3 °C above pre-industrial) there is more rainfall at all warming levels in CMIP6 (Fig. 7). Thus, the faster rate of global warming, the greater magnitude of Arctic amplification and the greater sensitivity of precipitation to warming likely all contribute to the more pronounced precipitation changes in CMIP6 than CMIP5.

Not surprisingly, sea-ice area declines faster in CMIP6 than CMIP5, leading to larger areas of open water (Fig. 6a). At the end of the century (relative to the start of the century), CMIP6 projects nearly twice as much open water in winter (around 9 million km$^2$) in the multi-model ensemble mean compared to 5.5 million km$^2$ in CMIP5 (Fig. 6a), resulting in a larger Arctic moisture source. In other seasons, CMIP6 also projects greater amounts of open water. As with temperature increase, the magnitude of the increase in open water area at the end of the

century across the models in both CMIPs is significantly correlated (r-values between 0.53 and 0.92, Supplementary Table 3) with the magnitude of the increase in rainfall (in all seasons) and with the magnitude of the snowfall decrease in summer and autumn (around −0.99 and −0.76, respectively) (Fig. 5b, e, h, k and Supplementary Table 3). Both CMIPs also project an increase in the vertically integrated horizontal moisture flux (VIMF) in all seasons, albeit larger in CMIP6 (Fig. 6c). In both CMIP ensembles, the largest increase occurs in summer (June–August), when moisture transport is climatologically the strongest[2]. However, this increase is stronger in CMIP6. The other three seasons show less distinction, particularly in spring, but nonetheless moisture transport changes are overall greater in CMIP6. As with temperature and open water, greater VIMF at the end of the century is significantly correlated with the stronger increase in rainfall in CMIP6 all seasons and with the stronger decreases in snowfall in summer and autumn (Fig. 5c, f, i, l and Supplementary Table 3).

**Transition to a rain-dominated precipitation.** As a result of the above-mentioned differences, the transition from a snow- to rain-dominated precipitation regime occurs earlier in CMIP6, particularly in autumn (Fig. 8j, k), with most of the Arctic Ocean, Siberia and the Canadian Archipelago becoming rainfall dominated one or two decades earlier (Fig. 8l). CMIP6 also shows a more rapid reduction of the snowfall ratio, from around 0.7 at the start of the century for both CMIPs to around 0.3 compared to 0.4 in CMIP5 by the end of the century

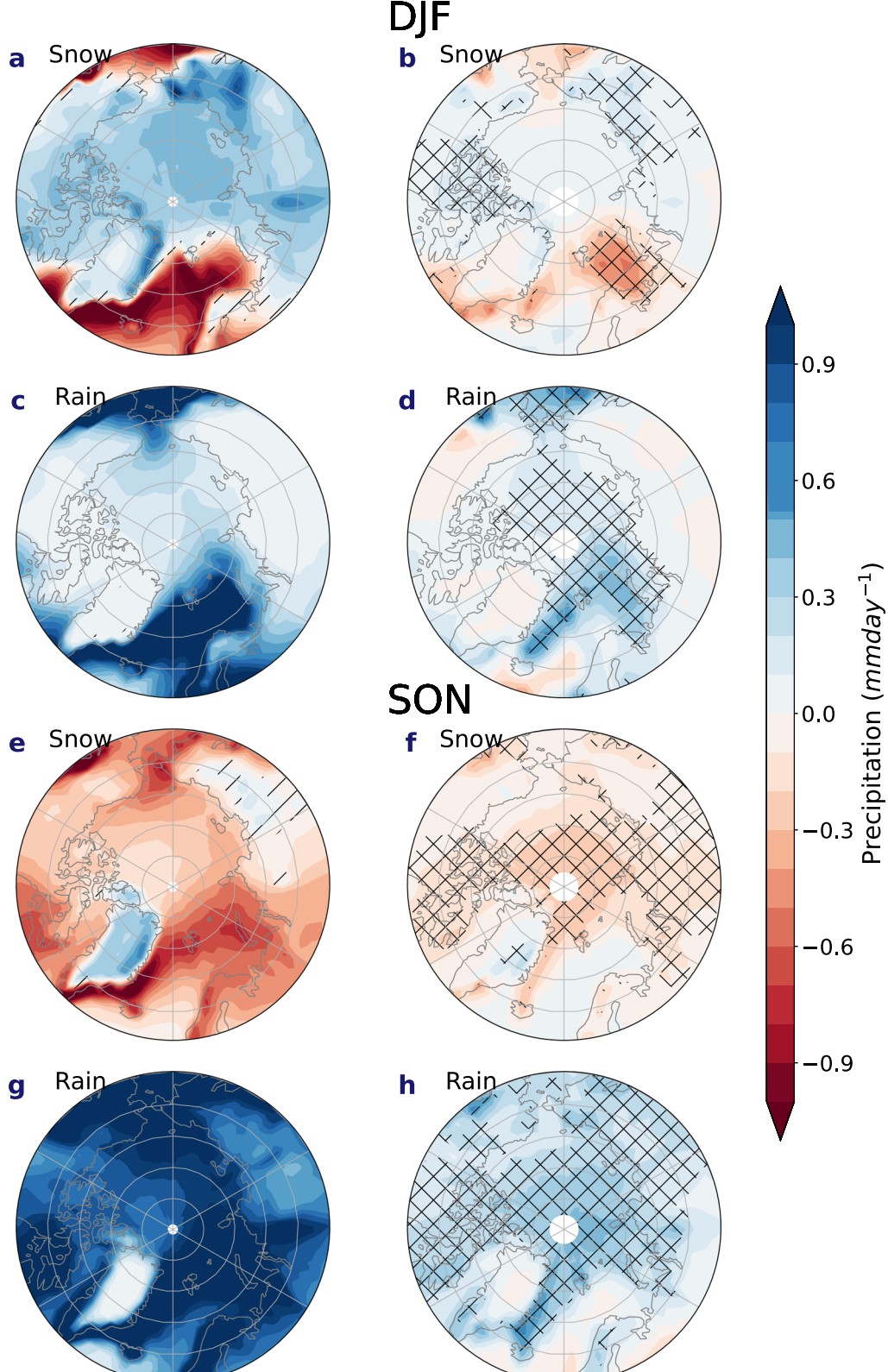

**Fig. 3 Snowfall and rainfall changes in CMIP6 and difference between CMIP6 and CMIP5 snowfall and rainfall at the end of the century.** The left-hand column shows the changes in **a**, **e** snowfall and **c**, **g** rainfall at the end of the century in **a**, **c** December–February (DJF) and **e**, **g** September–November (SON) in CMIP6. Straight line hatching indicates regions where differences are not statistically significant at the 95% confidence level. The right-hand column shows the difference in **b**, **f** snowfall and **d**, **h** rainfall at the end of the century (2091–2100) relative to the start of the century (2005–2014) between CMIP5 and CMIP6 (CMIP6–CMIP5) for **b**, **d** December–February and **f**, **h** September–November. Hatching indicates statistical significance at 95% confidence level.

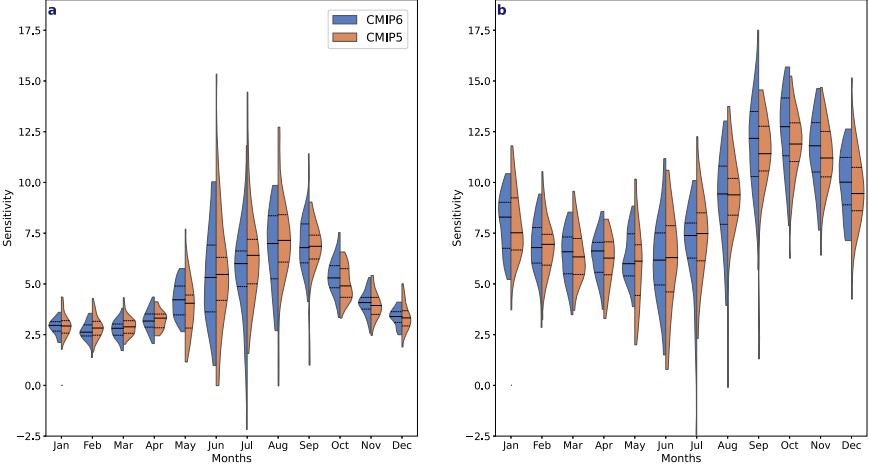

**Fig. 4 Sensitivity of precipitation to temperature change throughout the twenty-first century.** The percentage change of total precipitation per degree of **a** Arctic warming and **b** Global warming by month. The black dashed lines represent the 25th–75th percentile and the black solid lines represent the model mean following the SSP5–8.5/RCP8.5 scenarios for CMIP6 (blue) and CMIP5 (orange).

**Fig. 5 Model-dependent end-of-century changes in snowfall and rainfall for SSP5–8.5/RCP8.5 forcing.** Inter-model dependence of snowfall and rainfall to **a**, **d**, **g**, **j** surface air temperature, **b**, **e**, **h**, **k** Open water and **c**, **f**, **i**, **l** vertically integrated moisture flux (VIMF) changes at the end of the century relative to the start of the century for **a–c** December–February (DJF), **d–f** March–May (MAM), **g–i** June–August (JJA), **j–l** September–November (SON). Each dot represents a model from CMIP6 and each x represents a model from CMIP5. The stars represent the multi-model means for either CMIP6 and CMIP5 snowfall/rainfall in the same colours as their representative individual members. The line of best fit is calculated using a 1D polynomial.

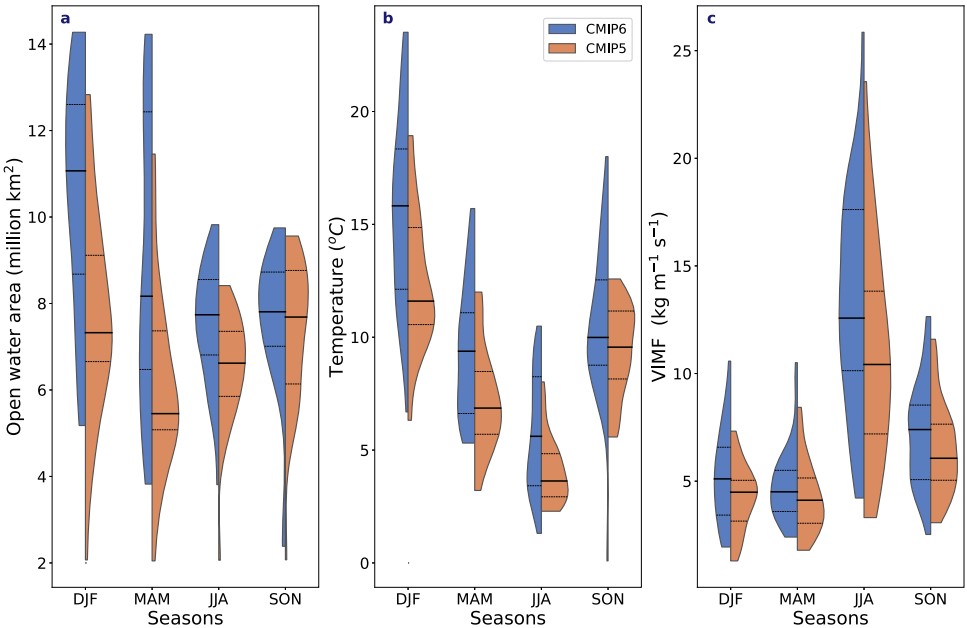

**Fig. 6 End-of-century changes in sea-ice concentration, temperature and vertically integrated moisture flux.** The end-of-century (2090–2100) changes relative to the start of the century (2005–2014) in all seasons in **a** open water area (million squared km), **b** temperature (°C) and **c** vertically integrated moisture flux (VIMF) at 70°N (kg m$^{-1}$ s$^{-1}$) for CMIP6 (blue) and CMIP5 (orange), where the colours represent the kernel distribution of all models. The solid black lines represent the multi-model ensemble means for both experiments and the dashed black lines denote the 25th and 75th percentile in each experiment.

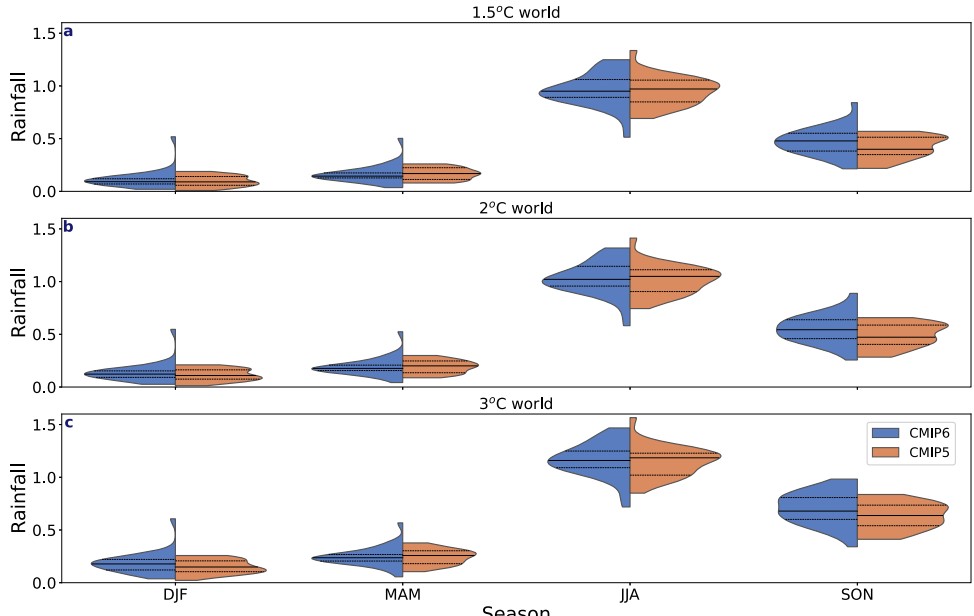

**Fig. 7 Arctic rainfall changes per season relative to global warming scenarios.** The Arctic rainfall change in mm day$^{-1}$ with respect to 1.5 °C, 2 °C and 3 °C global warming for CMIP6 (blue) and CMIP5 (orange) in December–February (DJF), March–May (MAM), June–August (JJA) and September–November (SON). Dashed black lines show the 25–75th percentiles and the solid black lines show the multi-model mean.

(Fig. 9d), indicating a more rapid loss of snowfall in CMIP6. By contrast, and despite large reductions in the snowfall-to-precipitation ratio, most of the Arctic remains largely snow-dominated in winter and spring throughout the century (Figs. 8a, b, d, e and 9a, b). Regionally, however, a transition to a rainfall regime in both winter and spring occurs ~10 years earlier in the Barents Sea in CMIP6 (Fig. 8c, f), but transitions later than CMIP5 in parts of North America and Europe in spring. This later transition in spring is likely due to the higher

snowfall ratio in CMIP6 at the start of the century (Fig. 9b). Although the snowfall ratio remains above 0.5 in winter throughout this century, the winter snowfall ratio, similar to autumn, still declines more rapidly in CMIP6 than CMIP5 at around 0.85 for both CMIP ensembles at the start of the century to around 0.75 and 0.8 for CMIP6 and CMIP5, respectively (Fig. 9a). In summer, the Arctic is largely rainfall dominated in both modelling experiments (Figs. 8g, h and 9c), except for north of 80°N.

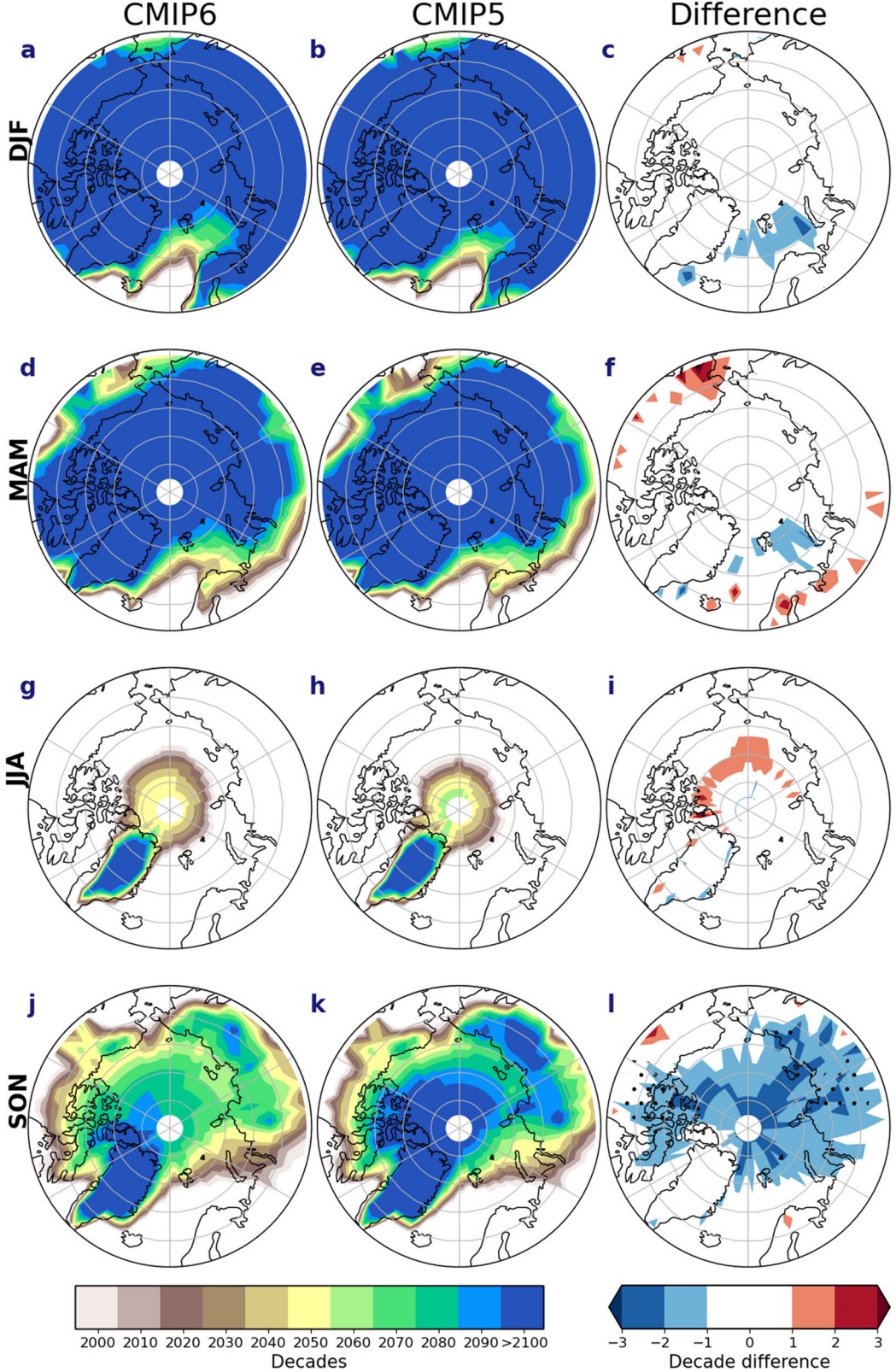

**Fig. 8 Decade of transition between snow-dominated precipitation to rainfall-dominated precipitation and their differences.** The decade of transition from a snow-dominated precipitation regime to a rain-dominated regime for CMIP6 [first column] and CMIP5 [second column] multi-model ensemble means, taken as when annual snowfall relative to annual precipitation falls below 50% and their differences [third column] for **a–c** December–February (DJF), **d–f** March–May (MAM), **g–i** June–August (JJA) and **j–l** September–November (SON). Areas that do not transition by 2100 are shaded in blue. Areas shaded in white in the first two columns are rain-dominated before the year 2000.

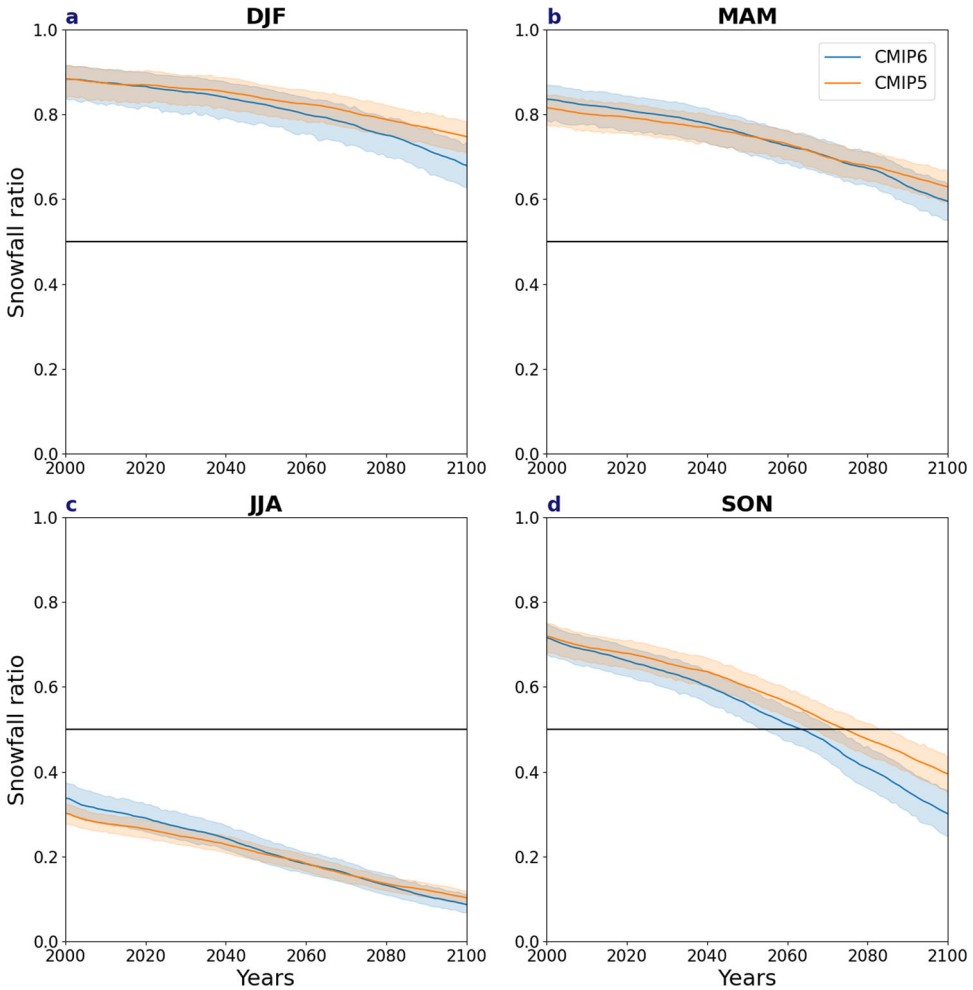

**Fig. 9 Time evolution of the snowfall ratio.** The snowfall ratio over time. The horizontal lines denote the transition from a snow- to rain-dominated precipitation at (0.5) for CMIP6 (blue) and CMIP5 (orange) for **a** December–February (DJF), **b** March–May (MAM), **c** June–August (JJA), **d** September–November (SON). The shading indicates 5th–95th model spread for each and all data are averaged as a 10-year rolling mean.

**Projected precipitation change under global warming scenarios.** Given the argued necessity to stay within the limits of 1.5 °C to 2 °C global warming to mitigate against severe climate change[34], it is useful to explore how these temperature limits relate to the transition to a rainfall-dominated precipitation regime across the Arctic. Assessing across different regions annually, the Beaufort, Chukchi, Bering, Laptev and East Siberian Seas remain snowfall dominated in both a 1.5 °C and 2 °C warmed world (Fig. 10), whereas transition to a rainfall-dominated precipitation regime will likely occur in the Greenland and Norwegian Seas regardless of a 1.5 °C or 2 °C limit, particularly in CMIP6. In western Russia and Europe, the transition to rainfall is more likely to require a 2 °C global warming, with more CMIP6 models showing this shift than CMIP5. Greenland is expected to transition to rainfall-dominated precipitation with 1.5 °C warming in CMIP6 only, but in both models when a 2 °C warming is realized. We also analysed the snowfall ratio for 3 °C global warming, as the likelihood of staying under a 2 °C warming is believed to be only around 5%[35] given current policies. At 3 °C warming, most regions, except those on the Pacific side of the Arctic have transitioned to a rainfall-dominated regime. However, seasonally, as indicated in Fig. 10, winter remains snowfall dominated by the end of the century and at 3 °C global warming most regions in the Arctic remain snow-dominated in winter and spring (Fig. 10).

## Discussion

Our analysis points to larger increases in Arctic precipitation in the CMIP6 projections compared to CMIP5, with the shift to an annually rain-dominated precipitation regime occurring approximately one or two decades earlier, with the greatest changes expected in autumn. The earlier shift towards a rain-dominated regime in CMIP6 has implications for the stability of social-ecological systems in the Arctic and the rate at which system changes will occur. This includes a further reduction in snow cover duration[8], which influences seasonality[34], ecosystem processes such as tundra greening[36], wildlife populations and human livelihoods[18,37]. Reduced snow cover will further exacerbate Arctic and global warming through albedo feedbacks[38], increased winter $CO_2$ fluxes[39] and methane releases[40] from soil[41] and thawing permafrost. These will additionally affect soil moisture and groundwater, influencing flood risk[42]. The transition to more rainfall will also impact the frequency of rain-on-snow (ROS) events, which can be devastating to wild caribou, reindeer and muskoxen populations[18,21,43], and result in a decline of fungal life[44]. Massive mortality following major ROS events has important social-ecological, cultural and economic implications[18,21]. Not all impacts will be negative, however. For example, the population of migratory birds has significantly increased due to a warmer and wetter Arctic[45].

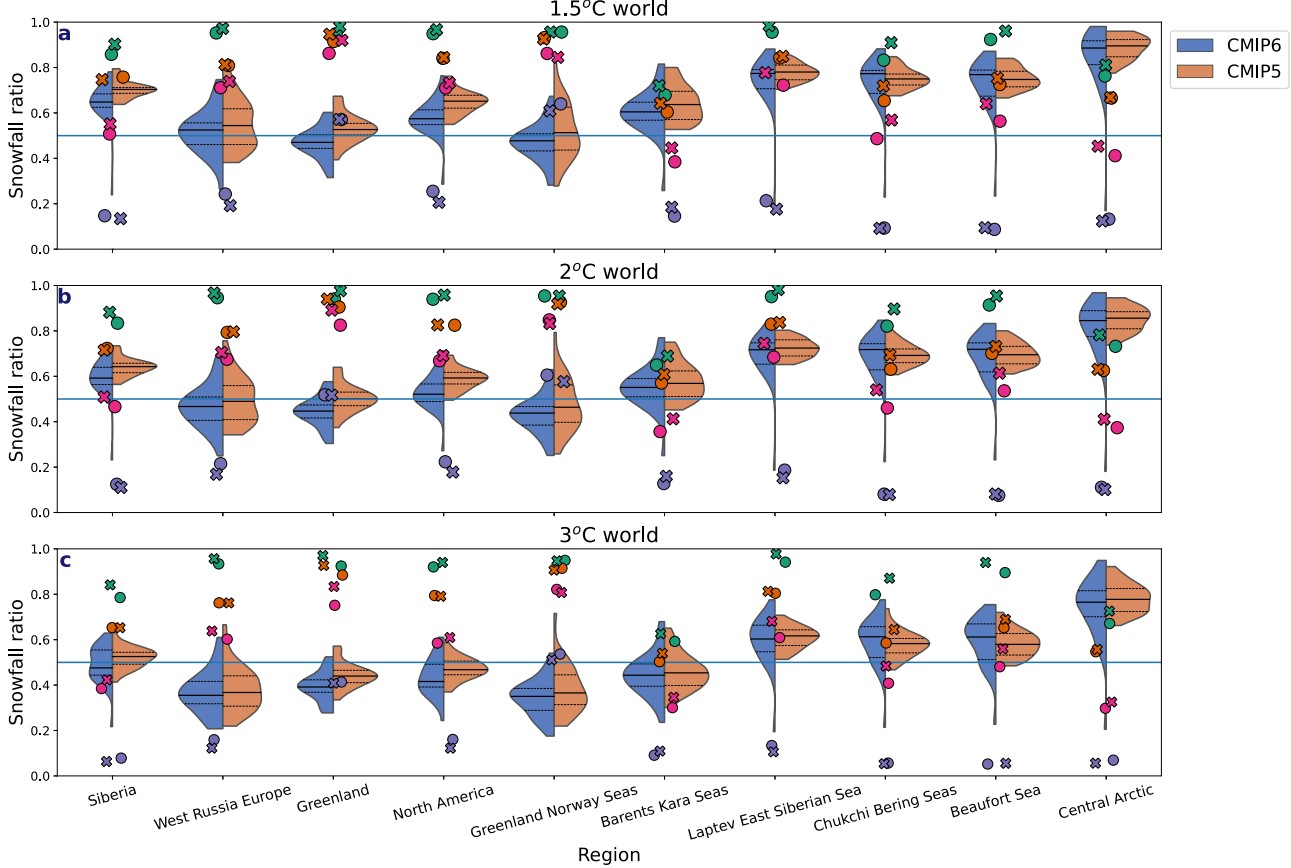

**Fig. 10 Snowfall ratio per region in a 1.5 °C, 2 °C and 3 °C global warming scenarios.** The snowfall ratio for different regions (identified in Supplementary Fig. 4) across the Arctic following 1.5 °C, 2 °C and 3 °C global warming for CMIP6 (blue) and CMIP5 (orange) with the dashed black lines showing the 25th–75th and the solid black lines showing the multi-model mean. The solid blue line identifies when the snowfall ratio is at 0.5. The circles and x's represent the multi-model ensemble mean snowfall ratio at 1.5 °C, 2 °C and 3 °C for CMIP6 (circles) and CMIP5 (x's) for December–February (green), March–May (orange), June–August (purple) and September–November (pink).

Increased precipitation may lead to increased river discharge[11], hastening sea-ice loss[46] and freshening of the ocean surface[47–49]. However, streamflow reductions have also been found following a transition to a rainfall-dominated precipitation regime due to greater evaporation from atmospheric warming[19]. Nevertheless, a change in either direction could negatively impact water availability downstream, resulting in flooding events that impact infrastructure such as roads and railways[50,51], and thus has implications for local communities[19]. Changes in precipitation over sea-ice areas will alter thermodynamic ice growth and snow depth; hence, the amount of light reaching the underside of the ice[52–54], impacting ocean stratification, circulation and ocean primary productivity[55]. Changes in light transmission, altering phytoplankton and algae blooms, will have cascading impacts through the marine food web[56]. The projected increased snowfall over Greenland in CMIP6 may mitigate mass loss from increased melting, potentially stabilizing the central part of the ice sheet. However, CMIP6 also projects greater rainfall around the southern and coastal edges of the ice sheet, which may destabilize these regions and accelerate Greenland's contribution to sea-level rise[16,57]. Greenland melt has already been shown to be at least twice the rate in CMIP6 than CMIP5[58] under the same radiative forcing.

We have shown that hydrological changes in the Arctic are amplified and the transition to a rain-dominated precipitation is expected to occur earlier and at a lower level of global warming in CMIP6 relative to CMIP5. This regime shift implies that more stringent mitigation policies are required as precipitation changes

which were expected with 2 °C of global warming above pre-industrial, now appear possible under 1.5 °C of global warming[18,21,36,37].

## Methods

**Model data sets.** Total precipitation, snowfall and rainfall (derived as the difference between total precipitation and snowfall) rates, open water (calculated based on the inverse of the sea-ice concentration), surface air temperature and the VIMF (using humidity and meridional component of the wind, details below) were analysed for both the CMIP5[59] and CMIP6[60] global climate models. These spanned the historical periods of 1960–2005 for CMIP5 and 1960–2014 for CMIP6, and then followed the highest emissions scenario, RCP8.5 from 2006 to 2100 for CMIP5 and SSP5–8.5 in 2015–2100 for CMIP6. SSP5–8.5 is designed to be similar to the RCP8.5 from CMIP5[61] (i.e., the radiative forcing at the end of the century remains at 8.5 W/m$^2$). Although RCP8.5 is a high-end emissions scenario, $CO_2$ cumulative emissions are presently tracking along the RCP8.5 scenario[62]. One ensemble member per model was used from both CMIP ensembles.

The Arctic region is defined here as the area north of 70°N as identified by the black line in Fig. 1c–f and statistical analyses were performed for this area only. Only models with simulations for both the historical and SSP/RCP scenarios in both CMIP6 and CMIP5 were included. No additional model pre-selection was employed.

End-of-century changes (taken as the average of 2091–2100), relative to the start of the century (the 10-year mean of 2005–2014, which for CMIP5 was conducted using the first 10 years of the RCP8.5 scenario), were computed for all analyzed variables. The transition decade was determined as the decade annual and seasonal snowfall-to-total precipitation falls to <<<50% as assessed from binning 10-year windows. The snowfall ratio over time in ERA5, CMIP6 and CMIP5 was calculated by dividing snowfall by total precipitation and averaged following a 10-year rolling mean.

The comparison between 1.5 °C, 2 °C and 3 °C global warming and the snowfall-to-precipitation ratio was based on global temperatures from 1960 to 2100 relative to the Intergovernmental Panel on Climate Change (IPCC) baseline of 1850–1900 and snowfall ratio from 1960 to 2100 for each model individually. This

was calculated using output from the RCP8.5 scenario. The actual scenario used should not influence the relative change of snowfall ratio to temperature. To calculate change in the snowfall-to-precipitation ratio relative to global temperature anomalies, which has a fairly linear and significant relationship[63], we regressed the global temperature anomalies against the snowfall-to-precipitation ratio and used the slope and intercept values to calculate what the ratio would be given a 1.5 °C, 2 °C and 3 °C temperature increase.

**Vertically integrated moisture flux**. The VIMF is:

$$f(\lambda, t) = \frac{1}{g} \int_{p_t}^{p_s} vq\,dp \qquad (1)$$

where $g$ is the acceleration due to gravity, $p_s$ is surface pressure which was taken at 1000 hPa, $p_t$ is the pressure level at 400 hPa above which humidity is negligible[10], $v$ is the meridional component of the wind and $q$ is specific humidity. Vertical heights were calculated using the trapezoidal rule. Statistical significance was evaluated based on the Student's $T$-test for differences between the end of century and start of century.

**Observational data**. For estimates of historical precipitation, we used data from both the ERA5 reanalysis[25] and the GPCP[26]. Means were computed for the period of overlap, 1979–2005. ERA5 is the fifth and latest global atmospheric reanalysis project from the European Center for Medium Range Weather Forecast covering the period 1979 to present. Outputs have a 31 km horizontal resolution and 137 levels in the vertical from the surface to 80 km in height. ERA5 was chosen following comparisons between different available reanalyses revealing that ERA5 had one of the best representations of precipitation relative to observations[64].

The GPCP is a monthly gridded precipitation data set, available at 2.5° × 2.5° grid resolution, is based on gauge observations and various satellite retrievals. GPCP has also been shown to perform poorly for high-latitude precipitation relative to observations[27] and reanalysis[28].

## Data availability

Climate model outputs from CMIP6 and CMIP5 are publicly available at https://esgf-node.llnl.gov/search/cmip6/ and https://esgf-node.llnl.gov/search/cmip5/. ERA5 reanalysis data and GPCP data are available at https://cds.climate.copernicus.eu/cdsapp#!/home and https://climatedataguide.ucar.edu/climate-data/gpcp-monthly-global-precipitation-climatology-project, respectively.

## Code availability

Code is available upon request to corresponding author.

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

## Acknowledgements
This research was undertaken, in part, thanks to funding from the Canada 150 Research Chairs Program (M.R.M. and J.S.), from NSF Grant NNA 198230 (M.S. and J.S.) and from the European Commission Research and Innovation Action Number 869471 (CHARTER) (B.F. and J.S.) and by NERC grant NE/V005855/1 (J.A.S.).

## Author contributions
M.R.M. and J.S. developed the ideas that led to this paper. M.R.M. performed the analysis of the data, created the figures and wrote the paper with input from J.S., M.S., J.A.S. and B.F.

## Competing interests
The authors declare no competing interests.
