## [Peer Review File · Nature Communications]

REVIEWER COMMENTS

Reviewer #1 (Remarks to the Author):

New model projections (CMIP6) indicate that Arctic sea ice loss may be more rapid than previously projected, and this study examines the resultant effects on Arctic precipitation. The new model, which the authors argue provides an improved simulation of the hydrologic cycle, projects large increases in total precipitation and especially rainfall, as previously known, but decades earlier than projected by earlier models. This effect is derived from the stronger projected increases in Arctic air temperature, and greater loss of sea ice (9 million versus 5.5 million km² in CMIP5), providing a much larger moisture source. The article examines regional differences and shows that while some areas remain snow-dominated at a global temperature rise of 1.5°C, almost all are rainfall dominated at +3°C.

Overall, the paper is clearly written, the methods adequately described and the figures present a large amount of information, including statistical analyses, in a compact, accessible form.

The results will be of broad scientific interest and impact. They also have major policy implications and add further urgency to decarbonization of the global economy, and to the development of adaptive management strategies.

The section on impacts is especially important, but focuses mostly on geophysical and ecological effects. The projected earlier, largescale changes in precipitation will have enormous impacts on built infrastructure in the North, where flooding is already of great concern, and some mention could be made of the literature on this; e.g. Instanes, A. et al. 2016. Changes to freshwater systems affecting Arctic infrastructure and natural resources. *J. Geophys. Res. Biogeosci.* 121: 567-85; Arp, C.D. et al. 2020. Recurring outburst floods from drained lakes: an emerging Arctic hazard. *Front. Ecol. Environ.* 18: 384-90.

The focus in this article is on the shift to 'rain dominance', which has a certain intuitive appeal as a concept, but does not mean a lot unless it is defined in absolute terms – it could just be a shift from 49% to 51% rainfall. Of course it is much more than that, and the initial projections in the opening paragraph of the article are especially compelling. It would be useful in the text (and even abstract) to refer to the % increase in rainfall by the end of this century as projected by this new model – in absolute % terms, as well as percent increment relative to CMIP5 (as is currently presented, and especially striking for SON in Fig. 4).

Minor comments:

Line 62: "Annual-mean precipitation from ERA5 is relatively similar to CMIP5 and CMIP6, lending confidence that both sets of model experiments do a reasonable job at reproducing Arctic precipitation."

Could a more exact statement be made than 'relatively similar... do reasonable job'?

Concerning this comparison, there are few climate stations in the Arctic region relative to the rest of the world. Apart from the inherent difficulties of measuring snowfall (as mentioned), how accurate is the ERA5 reanalysis for the North Polar region? (see below also).

Line 67: "We also compare the snowfall ratio (snowfall/precipitation)"

This is defined differently, erroneously I think, in supplement Figure S1: "Annual snowfall ratio (difference between snowfall and total precipitation)"

Line 231: "The transition decade from a snowfall to rainfall regime was found through determining in what decade annual and seasonal snowfall to total precipitation ratio fell to less than 50% through binning 10-year windows."

Simplify for clarity. The transition decade was identified as...

Line 243: "we regressed the global temperature anomalies against the to ratio and used the slope

and intercept values to"
Missing words - "snowfall to precipitation"

"Vertically integrated moisture flux."
Symbols were missing from the equation and text in my version of the manuscript.

Line 262: "ERA5 had one of the best representations of precipitation relative to observations"
Is this also true for the data-poor Arctic region?

Line 274: "All data are also available from the corresponding author upon reasonable request." Is this unusual qualifier necessary?

Reviewer #3 (Remarks to the Author):

This manuscript describes the use of reanalysis and CMIP5/6 climate model simulations to identify projected changes to Arctic precipitation amount and proportional phase (rainfall versus snowfall). The strength of the paper is the analysis of the CMIP output. The depth of analysis for the historical period is comparatively limited, but to a large extent this is influenced by the lack of high quality observational products for Arctic precipitation. The most notable new insights are:
-CMIP6 models project a transition of the precipitation regime to rain-dominant for more seasons, more regions, and earlier in the future compared to CMIP5;
-greater moisture transport into the Arctic in CMIP6 compared to CMIP5 output;
-projected precipitation changes are tied to the warming targets of the Paris Agreement.

While more detailed diagnosis of CMIP6 model performance for precipitation is no doubt still to be done, this paper provides a robust first determination of projected Arctic precipitation changes.

My comments are fairly limited, but I have some suggestions to improve the manuscript:

1. Lines 62-64: "Annual-mean precipitation from ERA5 is relatively similar to CMIP5 and CMIP6, lending confidence that both sets of model experiments do a reasonable job at reproducing Arctic precipitation."

To increase confidence in this statement, can you add a panel to Fig. S1 which shows the spatial pattern of ERA5 (and perhaps also GPCP) annual mean precipitation compared to CMIP5 and CMIP6? When integrated over the entire Arctic the CMIP and ERA5 time series agree reasonably well (as shown in Fig. S1), but there could be important regional differences with offsetting biases.

2. To ensure reproducibility, I suggest adding a table to the supplementary material which provides a list of the models which met your criteria ("Only models with simulations for both the historical and SSP/RCP scenarios in both CMIP6 and CMIP5 were included.") Also specify the realizations which were used...assume this to be r1 for each model?

3. The map figures include data to 60N, while the areally integrated analysis was cutoff at 70N. Was any analysis conducted using a consistent 60N cutoff, and if so what is the impact? Some regions with large projected precipitation changes are over the marginal sea ice zones south of 70N, and much of the Arctic land area is excluded when 70N is used as the southern limit.

4. In a number of places I think messaging needs to be clarified so as not to be confusing or over-simplified.:

Line 75: "Rainfall dominates the increase regardless of season."

Line 144: "By contrast, most of the Arctic remains largely snow-dominated in winter and spring through the end of the century"

The statement in line 75 refers to the proportional increase (by phase) of the mean total precipitation. If my understanding is correct, the winter precipitation increase in CMIP6 is driven by an increase in rainfall, not snowfall. But as noted on line 144, the winter season precipitation regime in the Arctic will still be snowfall dominated overall. Similarly, the statement of line 167 is an important finding (“...most regions, except those on the Pacific side of the Arctic have transitioned to a rainfall-dominated regime...”) but the seasonality of the changes is not captured with this wording. I suggest clarifying the wording in these places so that it is clear that most of the Arctic in winter will remain snowfall dominated in winter and spring at end of century as shown in Figure 4.

Similarly text on line 172 (“with the shift to a rain-dominated precipitation regime occurring decades earlier.”) should be clarified to be more specific on the seasonality.

5. Overall, the figures are very effective. But there is a heavy reliance on the supplementary material. For the reader interested in scenario dependence, precipitation sensitivity to temperature, and a regional perspective, all the key results are presented in the supplementary figures. I’m not sure what the editorial preference is, but this may require some restructuring.

Editorial comments

Line 31: This is a small point, but what about increased evapotranspiration from land?

Line 59: atmospheric sounders are also on satellites so just clean up this wording.

Line 90: change ‘implying’ to ‘consistent with’

Line 116: change ‘forecasted’ to ‘projected’

Line 127: change ‘open area’ to ‘open water area’

Lines 205-208: I understand the intended meaning but this sentence reads awkwardly to me. Consider rephrasing...

Line 265: another small point but the statement “GPCP has also been shown to perform poorly for high-latitude precipitation.” is based on a study from ~15 years ago. Has there been a more recent assessment?

Reviewer #4 (Remarks to the Author):

The manuscript presents an analysis on Arctic precipitation projections from the sixth phase of the Coupled Model Intercomparison Project (CMIP6). The analysis is straight-forward: it gives a very brief comparison of CMIP5 and CMIP6 precipitation with ERA-5 reanalysis and observations, shows the magnitude and rate of change in precipitation and precipitation phase for ~1960-2100 from CMIP5 and CMIP6, and includes projected changes in open water area, temperature, and moisture flux. In general, the results are a useful update in relation to previously published CMIP5 results, but I’m afraid the story of the manuscript is not wholly compelling. The results are consistent with other publications showing that the Arctic is projected to become a rain-dominated system under a warming climate. The biggest change from previous findings is the rate and magnitude of change, but even then, the analysis stops short of showing, in a statistically robust way, why the differences arise. The connections to possible causes (e.g. open water area increase, air temperature, etc.) are not statistically analyzed, but could be. Please see suggestions below for strengthening the manuscript.

Line 1. “than” not “then.”

Line 22. Is there a word missing in this sentence?

Lines 22-25. This needs to be revised for clarity.

Lines 51-53. I suggest rewording this last sentence. It sounds as if this study examined these impacts when it did not.

Lines 55-64. This is a weak assessment by only looking at annual averages over the entire Arctic. It would be tremendously more informative by taking it a step further: How do the spatial patterns compare between CMIP5, CMIP6, GPCP, and ERA5? What are the results for individual CMIP6 models rather than the inter-model mean?

Figures: How did you decide which figures should be supplementary vs. in the main text? Figure S1 would be helpful in the main text. Also, the white space in Figure S1 could be reduced.

Lines 58-59. References are needed. Also, do you mean under-catchment?

Lines 59-60. By how much? Please be more quantitative with the results described here and throughout this section.

Lines 61-62. It's tenuous to state this based on one discrepancy with ERA5. This statement needs more support through more thorough comparisons, references of related studies, or both.

Lines 62-64. This is vague. Can you please give numbers on their differences to give a better understanding of how they compare?

Lines 64-67. 0.1 mm per day is quite small. Is there an uncertainty or standard deviation that can be given here to get a better idea of their differences?

Line 68. What is the observational record? Please state the range in years here for clarity.

Lines 69-70. It would be easier to interpret these results if they were given as ratio change per decade.

Line 74. Please reword for clarity.

Lines 86-87. This is unclear and needs to be revised. First, it was already mentioned above that the CMIP6 models project larger increases in precipitation. Second, larger inter-model spread is an informative result worth expanding on. It would be useful to see the results from all models, perhaps as a bar chart or similar. Does the larger inter-model spread mean that there is more model disagreement when it comes to simulating future precipitation regimes? Can you provide more information (i.e. refer to other studies) that may help explain possible causes for the larger inter-model spread?

Line 94. Changes in what?

Line 105. What does "these" refer to?

Line 111. Do you mean "and" instead of "or" here? In the text, can you please define what you mean by sensitivity?

Lines 118-120. I suggest saying "attribute" rather than conclude. It's a subtle difference, but important since a statistical analysis wasn't performed on the two variables.

Line 128. Is there a typo here?

Lines 128-137. Again, this part of the analysis could be strengthened by conducting a statistical analysis of variables.

Line 144. Was the beginning ratio the same in CMIP5 as CMIP6. Please clarify in the text.

Lines 155-168. There needs to be a corresponding map for Figure S7 to show how regions were defined.

Line 173. This is vague. How many decades earlier?

Lines 242-245. Should we expect this geophysical relationship to be linear? Why or why not?

Line 247. Note, the equation and symbols in the following line do not show up in the pdf for reviewers.

Figure 1. It would be helpful to add y-axis tick labels on the right-hand side of the figure panels.

Figure 2. The dots of statistical significance are not visible with this color scheme. Please consider using a different color scheme or a different indication of statistical significance.

Figures 1-3. The font size of the axes labels and tick mark labels is very small. Please consider increasing the font size.

Figure 4. Should the color bar label for the difference figures be "days difference" rather than decades difference?

We thank all reviewers for their time to review this manuscript, and their positive feedback and constructive comments. We have incorporated these suggestions into the latest version of the manuscript paying particular attention to adding more statistical analysis, and incorporating more of the supplementary figures in the main manuscript. Our responses to all comments follow below in blue font.

Reviewer #1 (Remarks to the Author):

New model projections (CMIP6) indicate that Arctic sea ice loss may be more rapid than previously projected, and this study examines the resultant effects on Arctic precipitation. The new model, which the authors argue provides an improved simulation of the hydrologic cycle, projects large increases in total precipitation and especially rainfall, as previously known, but decades earlier than projected by earlier models. This effect is derived from the stronger projected increases in Arctic air temperature, and greater loss of sea ice (9 million versus 5.5 million km² in CMIP5), providing a much larger moisture source. The article examines regional differences and shows that while some areas remain snow-dominated at a global temperature rise of 1.5°C, almost all are rainfall dominated at +3°C.

Overall, the paper is clearly written, the methods adequately described and the figures present a large amount of information, including statistical analyses, in a compact, accessible form.

The results will be of broad scientific interest and impact. They also have major policy implications and add further urgency to decarbonization of the global economy, and to the development of adaptive management strategies.

We thank the reviewer for their time and supportive review.

The section on impacts is especially important, but focuses mostly on geophysical and ecological effects. The projected earlier, large-scale changes in precipitation will have enormous impacts on built infrastructure in the North, where flooding is already of great concern, and some mention could be made of the literature on this; e.g. Instanes, A. et al. 2016. Changes to freshwater systems affecting Arctic infrastructure and natural resources. *J. Geophys. Res. Biogeosci.* 121: 567-85; Arp, C.D. et al. 2020. Recurring outburst floods from drained lakes: an emerging Arctic hazard. *Front. Ecol. Environ.* 18: 384-90.

We agree, there are geophysical effects as well as ecological effects; and both are important. Thank you for bringing these papers to our attention. We have added details about these geophysical effects and cited the papers.

The focus in this article is on the shift to 'rain dominance', which has a certain intuitive appeal as a concept, but does not mean a lot unless it is defined in absolute terms – it could just be a shift from 49% to 51% rainfall. Of course it is much more than that, and the initial projections in the opening paragraph of the article are especially compelling. It would be useful in the text (and even abstract) to refer to the % increase in rainfall by the end of this century as projected by this

new model – in absolute % terms, as well as percent increment relative to CMIP5 (as is currently presented, and especially striking for SON in Fig. 4).

We thank the reviewer for this suggestion and have included the percentage increase in rainfall by the end of the century compared to 2020 in absolute values. There is a greater percentage increase in CMIP6 than in CMIP5, particularly in autumn when there is a 268% increase in total rainfall in CMIP6 compared to a 192% increase in CMIP5. We included this in the text at lines 93-95.

Minor comments:

Line 62: "Annual-mean precipitation from ERA5 is relatively similar to CMIP5 and CMIP6, lending confidence that both sets of model experiments do a reasonable job at reproducing Arctic precipitation."

Could a more exact statement be made than 'relatively similar... do reasonable job'?

These statements have been changed to make them more precise: in lines 73-75, we state that annual Arctic-mean precipitation in CMIP5 and CMIP6 is consistent with ERA5 and that both CMIP6 and CMIP5 have similar spatial variability in precipitation as ERA5 (as shown in figure 1).

Concerning this comparison, there are few climate stations in the Arctic region relative to the rest of the world. Apart from the inherent difficulties of measuring snowfall (as mentioned), how accurate is the ERA5 reanalysis for the North Polar region? (see below also).

As the reviewer correctly points out, there are limited climate stations in the Arctic region. Therefore, it is hard to fully determine the accuracy of ERA5 for the Arctic. Other products such as buoys were used by Wang et al. (2019), which compared surface temperature, total precipitation and snowfall from both ERA-Interim and ERA5 to buoy data. They found that there was more precipitation in ERA5, which reduced the global dry bias in ERA-Interim. They also found that, relative to the buoy data, ERA5 had a better representation of total precipitation and snowfall than ERA-Interim. Furthermore, although based on snowfall measurements, Cabaj et al. (2020) also found that ERA5 snowfall in summer matches CloudSAT snowfall quite well. This is discussed in lines 68-72.

Line 67: "We also compare the snowfall ratio (snowfall/precipitation)"

This is defined differently, erroneously I think, in supplement Figure S1: "Annual snowfall ratio (difference between snowfall and total precipitation)"

Thank you for pointing out this error. As noted, the snowfall ratio is calculated as ratio of snowfall to total precipitation and is not a difference. This has been changed in the Figure caption, which is now Figure 1 in the revised manuscript.

Line 231: “The transition decade from a snowfall to rainfall regime was found through determining in what decade annual and seasonal snowfall to total precipitation ratio fell to less than 50% through binning 10-year windows.”

Simplify for clarity. The transition decade was identified as...

We have simplified the sentence as suggested (line 288 in updated manuscript).

Line 243: “we regressed the global temperature anomalies against the to ratio and used the slope and intercept values to”

Missing words - “snowfall to precipitation”

Corrected.

“Vertically integrated moisture flux.”

Symbols were missing from the equation and text in my version of the manuscript.

Corrected.

Line 262: “ERA5 had one of the best representations of precipitation relative to observations”

Is this also true for the data-poor Arctic region?

Following the same issue raised earlier, Cabaj et al. (2020) find that ERA5 is has one of the best representations of precipitation relative to observations for the Arctic region, Cabaj et al (2020) also compared CloudSat snowfall measurements with ERA-Interim, ERA5 and MERRA2, found that ERA5 matches best with CloudSat in the summer, although ERA-Interim compared better to CloudSat in winter.

Line 274: “All data are also available from the corresponding author upon reasonable request.”

Is this unusual qualifier necessary?

We have removed this line.

Reviewer #3 (Remarks to the Author):

This manuscript describes the use of reanalysis and CMIP5/6 climate model simulations to identify projected changes to Arctic precipitation amount and proportional phase (rainfall versus snowfall). The strength of the paper is the analysis of the CMIP output. The depth of analysis for the historical period is comparatively limited, but to a large extent this is influenced by the lack of high quality observational products for Arctic precipitation. The most notable new insights are:

- CMIP6 models project a transition of the precipitation regime to rain-dominant for more seasons, more regions, and earlier in the future compared to CMIP5;
- greater moisture transport into the Arctic in CMIP6 compared to CMIP5 output;
- projected precipitation changes are tied to the warming targets of the Paris Agreement.

While more detailed diagnosis of CMIP6 model performance for precipitation is no doubt still to be done, this paper provides a robust first determination of projected Arctic precipitation changes.

We thank the reviewer for the time taken to complete this review, and the constructive and supportive comments.

My comments are fairly limited, but I have some suggestions to improve the manuscript:

1. Lines 62-64: "Annual-mean precipitation from ERA5 is relatively similar to CMIP5 and CMIP6, lending confidence that both sets of model experiments do a reasonable job at reproducing Arctic precipitation."

To increase confidence in this statement, can you add a panel to Fig. S1 which shows the spatial pattern of ERA5 (and perhaps also GPCP) annual mean precipitation compared to CMIP5 and CMIP6? When integrated over the entire Arctic the CMIP and ERA5 time series agree reasonably well (as shown in Fig. S1), but there could be important regional differences with offsetting biases.

Following this suggestion, we added additional plots and note that both CMIPs do a reasonable job and replicate the spatial distribution of precipitation relative to ERA5.

2. To ensure reproducibility, I suggest adding a table to the supplementary material which provides a list of the models which met your criteria ("Only models with simulations for both the historical and SSP/RCP scenarios in both CMIP6 and CMIP5 were included.") Also specify the realizations which were used...assume this to be r1 for each model?

This has been created and added to the supplemental material. We also amended the methods section to state that the first ensemble member was used per model.

3. The map figures include data to 60N, while the areally integrated analysis was cutoff at 70N. Was any analysis conducted using a consistent 60N cutoff, and if so what is the impact? Some

regions with large projected precipitation changes are over the marginal sea ice zones south of 70N, and much of the Arctic land area is excluded when 70N is used as the southern limit.

We analysed precipitation change poleward of 60°N and found that the results were essentially the same as those for the region poleward of 70°N. We decided to continue to use the definition of the Arctic as poleward of 70°N following other studies such as Bintanja and Andry (2017) who also use this definition. We added a bold line around 70°N in the spatial plots shown in Figure 1.

4. In a number of places I think messaging needs to be clarified so as not to be confusing or over-simplified.:

Line 75: "Rainfall dominates the increase regardless of season."

Line 144: "By contrast, most of the Arctic remains largely snow-dominated in winter and spring through the end of the century"

These lines have been edited for clarification.

The statement in line 75 refers to the proportional increase (by phase) of the mean total precipitation. If my understanding is correct, the winter precipitation increase in CMIP6 is driven by an increase in rainfall, not snowfall. But as noted on line 144, the winter season precipitation regime in the Arctic will still be snowfall dominated overall. Similarly, the statement of line 167 is an important finding ("...most regions, except those on the Pacific side of the Arctic have transitioned to a rainfall-dominated regime...") but the seasonality of the changes is not captured with this wording. I suggest clarifying the wording in these places so that it is clear that most of the Arctic in winter will remain snowfall dominated in winter and spring at end of century as shown in Figure 4.

Similarly text on line 172 ("with the shift to a rain-dominated precipitation regime occurring decades earlier.") should be clarified to be more specific on the seasonality.

We thank the reviewer for this comment and have edited the text throughout the manuscript to better reflect the seasonality of these changes.

5. Overall, the figures are very effective. But there is a heavy reliance on the supplementary material. For the reader interested in scenario dependence, precipitation sensitivity to temperature, and a regional perspective, all the key results are presented in the supplementary figures. I'm not sure what the editorial preference is, but this may require some restructuring.

We agree, as also raised by reviewer 4, we restructured the paper to include figures from the supplementary materials and also added new figures to answer points raised by the reviewers.

Editorial comments

Line 31: This is a small point, but what about increased evapotranspiration from land?

We added mention of evapotranspiration to this point, recognising that Vihma et al., (2016) noted a positive trend of evapotranspiration across the Arctic.

Line 59: atmospheric sounders are also on satellites so just clean up this wording.

Done. .

Line 90: change 'implying' to 'consistent with'

Done. .

Line 116: change 'forecasted' to 'projected'

Done.

Line 127: change 'open area' to 'open water area'

Done.

Lines 205-208: I understand the intended meaning but this sentence reads awkwardly to me. Consider rephrasing...

This line has been edited for clarity.

Line 265: another small point but the statement "GPCP has also been shown to perform poorly for high-latitude precipitation." is based on a study from ~15 years ago. Has there been a more recent assessment?

A more recent study (Marcivecchio et al., 2021) compared precipitation over the Arctic in GPCP, ERA-Interim, ERA5, and MERRA2, although not to any direct observations, and found that precipitation in GPCP is much lower than that of ERA-Interim and ERA5, which is what we also find in Figure S1 (now Figure 1 in the manuscript) with relation to ERA5. Given the studies on ERA5 compared to satellite and buoy data in the Arctic, which show a reasonable agreement in this reanalysis product to the observations, it might appear that, GPCP does still perform poorly for Arctic precipitation.

Reviewer #4 (Remarks to the Author):

The manuscript presents an analysis on Arctic precipitation projections from the sixth phase of the Coupled Model Intercomparison Project (CMIP6). The analysis is straight-forward: it gives a very brief comparison of CMIP5 and CMIP6 precipitation with ERA-5 reanalysis and observations, shows the magnitude and rate of change in precipitation and precipitation phase for ~1960-2100 from CMIP5 and CMIP6, and includes projected changes in open water area, temperature, and moisture flux. In general, the results are a useful update in relation to previously published CMIP5 results, but I'm afraid the story of the manuscript is not wholly compelling. The results are consistent with other publications showing that the Arctic is projected to become a rain-dominated system under a warming climate. The biggest change from previous findings is the rate and magnitude of change, but even then, the analysis stops short of showing, in a statistically robust way, why the differences arise. The connections to possible causes (e.g. open water area increase, air temperature, etc.) are not statistically analyzed, but could be. Please see suggestions below for strengthening the manuscript.

We thank the reviewer for their time taken to undertake this review and for providing such a detailed and constructive review.

Line 1. "than" not "then."

Changed.

Line 22. Is there a word missing in this sentence?

We have edited this line.

Lines 22-25. This needs to be revised for clarity.

This line has been edited.

Lines 51-53. I suggest rewording this last sentence. It sounds as if this study examined these impacts when it did not.

We edited this final sentence to ensure that there is no misinterpretation about what this study focuses on.

Lines 55-64. This is a weak assessment by only looking at annual averages over the entire Arctic. It would be tremendously more informative by taking it a step further: How do the spatial patterns compare between CMIP5, CMIP6, GPCP, and ERA5? What are the results for individual CMIP6 models rather than the inter-model mean?

In line with a comment from Reviewer 3 we added the spatial annual precipitation in all 4 datasets for comparison. We find that CMIPs capture most of the spatial variability of the reanalysis and GPCP although weaker, likely due to the averaging of over 30 models for each dataset. We also examined the historical time series for each model of CMIP6 relative to its multi-model-mean and find fairly large spread between models such as in 1979 with one model (FGOALS-g3) around 0.65mm day^{-1} relative to ACCESS-ESM1-5 which has precipitation at 1.15 mm day^{-1} .

Figures: How did you decide which figures should be supplementary vs. in the main text? Figure S1 would be helpful in the main text. Also, the white space in Figure S1 could be reduced.

We thank the reviewer for this comment, which is in line with the recommendations of the other reviewers. We moved four of the figures from the supplementary figures into the main text.

Lines 58-59. References are needed. Also, do you mean under-catchment?

Reference has been included here and the text edited.

Lines 59-60. By how much? Please be more quantitative with the results described here and throughout this section.

We added more qualitative analysis on the differences in the historical period precipitation.

Lines 61-62. It's tenuous to state this based on one discrepancy with ERA5. This statement needs more support through more thorough comparisons, references of related studies, or both.

As suggested by reviewer 3, we added the climatological spatial patterns of all datasets for a more complete comparison of ERA5 and GPCP. As included in the updated Figure 1 (which was moved from the supplemental figures), the spatial pattern of the climatological mean in GPCP is fairly similar to ERA5, as well as CMIP5 and CMIP6 for the 1979-2005 period. Nonetheless, a weak bias is still evident in GPCP in both this study and in others such as in Marcovecchio et al. (2021), which found lower precipitation rates and different seasonal variations to reanalysis, and Serreze et al. (2005) who found that reanalysis have a better representation of monthly precipitation than GPCP. We included this additional analysis in lines 62-68.

Lines 62-64. This is vague. Can you please give numbers on their differences to give a better understanding of how they compare?

As also suggested by Reviewer 1, we changed the wording to be more allow for a better comparison (see lines 72-75).

Lines 64-67. 0.1 mm per day is quite small. Is there an uncertainty or standard deviation that can be given here to get a better idea of their differences?

The standard deviation has been included.

Line 68. What is the observational record? Please state the range in years here for clarity.

Done.

Lines 69-70. It would be easier to interpret these results if they were given as ratio change per decade.

We have edited this to show the ratio change per decade.

Line 74. Please reword for clarity.

This has been rewritten.

Lines 86-87. This is unclear and needs to be revised. First, it was already mentioned above that the CMIP6 models project larger increases in precipitation. Second, larger inter-model spread is an informative result worth expanding on. It would be useful to see the results from all models, perhaps as a bar chart or similar. Does the larger inter-model spread mean that there is more model disagreement when it comes to simulating future precipitation regimes? Can you provide more information (i.e. refer to other studies) that may help explain possible causes for the larger inter-model spread?

We included two new tables in the supplementary section which show the mean, standard deviation, the 5th and 95th percentiles and the difference between these for the last ten years of the record (2091-2100) for each of the CMIPs, for each season and all precipitation types (total precipitation, snowfall and rainfall) (Table S1). We also made the same table for the surface air temperature, open water and vertically integrated moisture flux (VIMF) (Table S2). From Table S1, we found that there is a larger standard deviation in almost every precipitation type for each season in CMIP6 than CMIP5, including a larger mean. The overall spread is greater in CMIP6, with larger differences in the percentiles in each season and precipitation type. This also coincides with a larger intermodel spread in surface air temperature, open water and vertically integrated moisture flux (Table S2) for all seasons in CMIP6 relative to CMIP5. This analysis has been incorporated into the manuscript at lines 108-121.

Line 94. Changes in what?

We edited this to highlight that we are showing the spatial changes in snowfall and rainfall at the end of the century.

Line 105. What does “these” refer to?

This is in reference to the snowfall changes; we edited this line for greater clarity.

Line 111. Do you mean “and” instead of “or” here? In the text, can you please define what you mean by sensitivity?

We meant ‘or’ in this context and have changed the start of this sentence to clarify that we are assessing which of these changes may have resulted in the larger precipitation change in CMIP6. By sensitivity, we mean the percentage change in precipitation per degree of global warming; we have edited the text to more clearly represent this point.

Lines 118-120. I suggest saying “attribute” rather than conclude. It’s a subtle difference, but important since a statistical analysis wasn’t performed on the two variables.

Done.

Line 128. Is there a typo here?

Thank you for pointing this out, there was a word missing here as also noted by reviewer 1.

Lines 128-137. Again, this part of the analysis could be strengthened by conducting a statistical analysis of variables.

We added a new figure to highlight the relationship of snowfall and rainfall to surface air temperature, open water and vertically integrated moisture flux (Figure 5) and included a table of correlation coefficients (Table S3) to show the correlations of snowfall and rainfall in each season to each aforementioned variable. We found that in all seasons for each variable there is an increase in rainfall with increases in surface air temperature, vertically integrated moisture flux (VIMF) and open water which is much stronger in CMIP6 (due to greater increases in surface air temperatures, VIMF and open water) relative to CMIP5. Furthermore, these relationships are stronger in autumn than any other season. We furthermore show that these relationships in rainfall are significantly positively correlated in all seasons across almost all variables and negatively correlated for snowfall in most seasons. We included this additional analysis in lines 141-164 in the revised manuscript.

Line 144. Was the beginning ratio the same in CMIP5 as CMIP6. Please clarify in the text.

Thank you for this suggestion, we added text to clarify that the ratio in autumn and winter at the start of the century in both CMIPs was the same.

Lines 155-168. There needs to be a corresponding map for Figure S7 to show how regions were defined.

This has been created and added to the supplemental information.

Line 173. This is vague. How many decades earlier?

We have added more detail here.

Lines 242-245. Should we expect this geophysical relationship to be linear? Why or why not?

Yes, we expect a fairly linear relationship in line with Byun et al (2008) *A snowfall-ratio equation and its application to Numerical Snowfall prediction*, Weather and Forecasting, who found a significant dependence of snowfall ratio on the surface air temperature due to either higher temperatures resulting in more rainfall and/or melting snow as it falls. We added more detail about this relationship at lines 298-300.

Line 247. Note, the equation and symbols in the following line do not show up in the pdf for reviewers.

Corrected.

Figure 1. It would be helpful to add y-axis tick labels on the right-hand side of the figure panels.

Added.

Figure 2. The dots of statistical significance are not visible with this color scheme. Please consider using a different color scheme or a different indication of statistical significance.

Addressed. The figure is clearer now.

Figures 1-3. The font size of the axes labels and tick mark labels is very small. Please consider increasing the font size.

Font sizes of all have been increased.

Figure 4. Should the color bar label for the difference figures be “days difference” rather than decades difference?

We edited the text on the colour bar to reflect that this is either 1, 2, or 3 decades rather than the original which showed 10,20,30 which falsely indicated that changes were 10 decades different.

REVIEWERS' COMMENTS

Reviewer #1 (Remarks to the Author):

The authors have adequately responded to each of my review comments and I recommend acceptance for publication.

Reviewer #3 (Remarks to the Author):

Thanks to the authors for their substantive and careful revisions to the manuscript. My primary concerns raised in review of the initial submission were addressed through:

- addition of a new panel in Figure 1
- addition of tables in the supplementary material specifying the models used in the analysis
- a substantial shift of content from supplementary material to the main paper

I note also that the responses to suggestions from Reviewer 4 resulted in the generation of new results (e.g. Figure 5) which have also improved the manuscript.

The analysis provides a clear and robust assessment of projected Arctic precipitation changes in CMIP6 models compared to CMIP5. Given the extreme rainfall event observed at Greenland Summit this summer, this is timely analysis which will support future Arctic climate assessments. My remaining minor comments are below.

Line 17/18 and line 31: the term 'evapotranspiration' is employed, but only the dependence on open water is specified. So this should be changed to 'evaporation' or add a phrase to specify land processes as well.

Line 24: I think it is more precise to state "The transition from a snow- to rain-dominated Arctic in the summer and autumn is projected..."

Lines 49/50: Similarly, change to "...an earlier transition to a rainfall-dominated Arctic in the summer and autumn."

Line 59: this is incorrect as currently worded. Change to "...undercatch of snowfall."

Line 75: this should be mm day-1?

Lines 83, 90, 126, 152, 162, 182, 184, 206, 285: change from 'both CMIPs' to 'both CMIP ensembles'

Line 87: In this paragraph, the focus shifts from the historical period to climate model projections. The forcing scenarios (RCP 8.5 for CMIP5; SSP5-8.5 for CMIP6) should be specified in the main text, not just the caption to Figure 2. This will help clarify why RCP4.5 is explicitly mentioned in lines 108 and 124, which is currently confusing.

Line 123: instead of 'Larger spatial changes...' is it better to say 'More extensive spatial changes...?'

Lines 158/159: does this warming of 23C and 18C represent the maximum value from individual models within each ensemble? Perhaps make this clear...

Table S2: round off and remove decimals from mean open water values

Reviewer #4 (Remarks to the Author):

I commend the authors for taking the time to address my concerns. The manuscript has greatly improved. Below are minor suggestions. Well done on the nice paper.

- Legends in Figures 1, 2, 4, 6, 7, 10 are difficult to read being so small.
- The colors described in Table S2's caption are inconsistent with table color scheme.
- Please consider using a different color scheme to mitigate issues for red/green colorblindness; this is particularly problematic for figures 5 and 10 and in Table S3.

REVIEWERS' COMMENTS

We wish to express our appreciation to all reviewers for the work and time spent on reviewing this manuscript, their positive feedback and that they have found this manuscript suitable for publication. We have edited the manuscript and updated figures as suggested below as noted in blue font.

Reviewer #1 (Remarks to the Author):

The authors have adequately responded to each of my review comments and I recommend acceptance for publication.

Reviewer #3 (Remarks to the Author):

Thanks to the authors for their substantive and careful revisions to the manuscript. My primary concerns raised in review of the initial submission were addressed through:

- addition of a new panel in Figure 1
- addition of tables in the supplementary material specifying the models used in the analysis
- a substantial shift of content from supplementary material to the main paper

I note also that the responses to suggestions from Reviewer 4 resulted in the generation of new results (e.g. Figure 5) which have also improved the manuscript.

The analysis provides a clear and robust assessment of projected Arctic precipitation changes in CMIP6 models compared to CMIP5. Given the extreme rainfall event observed at Greenland Summit this summer, this is timely analysis which will support future Arctic climate assessments. My remaining minor comments are below.

We thank the reviewer for their further suggestions to improve the manuscript and have made the edits as suggested below

Line 17/18 and line 31: the term 'evapotranspiration' is employed, but only the dependence on open water is specified. So this should be changed to 'evaporation' or add a phrase to specify land processes as well.

We have removed the term evapotranspiration and simply included evaporation

Line 24: I think it is more precise to state “The transition from a snow- to rain-dominated Arctic in the summer and autumn is projected...”

Changed

Lines 49/50: Similarly, change to “...an earlier transition to a rainfall-dominated Arctic in the summer and autumn.”

Changed

Line 59: this is incorrect as currently worded. Change to “...undercatch of snowfall.”

Changed

Line 75: this should be mm day⁻¹?

Changed

Lines 83, 90, 126, 152, 162, 182, 184, 206, 285: change from ‘both CMIPs’ to ‘both CMIP ensembles’

Changed

Line 87: In this paragraph, the focus shifts from the historical period to climate model projections. The forcing scenarios (RCP 8.5 for CMIP5; SSP5-8.5 for CMIP6) should be specified in the main text, not just the caption to Figure 2. This will help clarify why RCP4.5 is explicitly mentioned in lines 108 and 124, which is currently confusing.

We have included here that this precipitation increase is in the RCP8.5/SSP5-8.5 scenarios.

Line 123: instead of ‘Larger spatial changes...’ is it better to say ‘More extensive spatial changes...’?

Changed

Lines 158/159: does this warming of 23C and 18C represent the maximum value

from individual models within each ensemble? Perhaps make this clear...

This correctly refers to the maximum value in an individual model and we have edited the text to better reflect this.

Table S2: round off and remove decimals from mean open water values

Changed

Reviewer #4 (Remarks to the Author):

I commend the authors for taking the time to address my concerns. The manuscript has greatly improved. Below are minor suggestions. Well done on the nice paper.

We thank the reviewer for their time to review this again and for their constructive comments and suggestions

- Legends in Figures 1, 2, 4, 6, 7, 10 are difficult to read being so small.

All legends have been increased in size

- The colors described in Table S2's caption are inconsistent with table color scheme.

Thank you for pointing this out, this has been changed

- Please consider using a different color scheme to mitigate issues for red/green colorblindness; this is particularly problematic for figures 5 and 10 and in Table S3.

Thank you for drawing our attention to this and we have corrected this using colour-blind friendly colour schemes